# Accelerated knowledge discovery from omics data by optimal experimental design

Xiaokang Wang[1,2], Navneet Rai[2,3], Beatriz Merchel Piovesan Pereira[2,4], Ameen Eetemadi[2,3] & Ilias Tagkopoulos [2,3✉]

How to design experiments that accelerate knowledge discovery on complex biological landscapes remains a tantalizing question. We present an optimal experimental design method (coined OPEX) to identify informative omics experiments using machine learning models for both experimental space exploration and model training. OPEX-guided exploration of *Escherichia coli*'s populations exposed to biocide and antibiotic combinations lead to more accurate predictive models of gene expression with 44% less data. Analysis of the proposed experiments shows that broad exploration of the experimental space followed by fine-tuning emerges as the optimal strategy. Additionally, analysis of the experimental data reveals 29 cases of cross-stress protection and 4 cases of cross-stress vulnerability. Further validation reveals the central role of chaperones, stress response proteins and transport pumps in cross-stress exposure. This work demonstrates how active learning can be used to guide omics data collection for training predictive models, making evidence-driven decisions and accelerating knowledge discovery in life sciences.

[1] Department of Biomedical Engineering, University of California, Davis, CA 95616, USA. [2] Genome Center, University of California, Davis, CA 95616, USA. [3] Department of Computer Science, University of California, Davis, CA 95616, USA. [4] Microbiology Graduate Group, University of California, Davis, CA 95616, USA. ✉email: itagkopoulos@ucdavis.edu

The scientific method of formulating, testing, and ultimately accepting hypotheses has been the way we have advanced science for centuries[1–3]. Through hypothesis-driven and discovery-based science, we have been able to study an organism's physiology intimately, fueled by technological advances in structural biology[4], omics[5], automation[6], computational modeling, and big data analysis[7,8]. In computational science, we are witnessing the age of predictive and prescriptive analytics in a wide spectrum of disciplines, including biology, biotechnology, and medicine, with large, consistent, and informative datasets being essential for computational learning[9–16].

Through human history, data are viewed as a means to test hypotheses or discover associations and phenomena, and less as a means to train computational methods across the various biological dimensions[17]. In the era of prediction, experimental design methods, such as those based on optimal experimental design (OED), also known as active learning, have the potential to accelerate scientific exploration[18–20]. OED is a class of algorithms originating from the 1950s that aims to guide data collection[21]. OED methods usually formulate a sampling problem as an optimization problem, which aims to identify the next experiment(s) to perform so that a specific objective is maximized, and a set of constraints are met[22]. These methods usually balance exploration (global search, maximizing coverage of the experimental space) and exploitation (local search, refining existing solutions) objectives. Depending on the problem, mutual information[23], Fisher information[24], predictive variance[25], generalization error[26,27], and margin[28] have been used as part of the objective function[29]. OED methods have been applied extensively in various industries, including aerospace engineering[30], seismic source inversion[31], sensor placement[32], and more recently in material science[22,33,34]. In biology, OED methods have recently been used in protein design[35], drug discovery[36–39], assay panel selection[40], system biology[41–47], and synthetic biology[48]. The systems biology applications of OED have been largely focused on uncovering the underlying gene regulatory or signaling network, usually involving a few dozen genes[49,50]. In contrast, our focus is on optimal training of predictive machine learning models from genome-scale transcriptional profiling experiments, which aim to capture the expression of thousands of genes. Although never used for omics experiments, OED methods can be especially useful in exploring the experimental space efficiently across a multitude of design dimensions and providing a method to produce training data that carry the maximum information content for training a predictive model.

Towards this goal, we design an optimal experimental design (coined OPEX) framework to guide omics experimentation by selecting the most informative experiments to perform. OPEX consists of two essential modules, a machine learning model and a utility metric that evaluates the information of an unobserved datapoint. Here we use Gaussian process (GP) as the model of gene expression[51], both because it is a nonparametric method and it predicts a distribution rather than a point estimate. From the predicted distribution, we use two metrics, entropy, and mutual information[32], to evaluate the utility of each candidate experiment.

In its core, OPEX assesses the information content distribution and model uncertainty across the experimental space to identify the next batch of experiments (Fig. 1). We apply OPEX in the exploration of the transcriptional interplay in *Escherichia coli* when exposed to biocides and antibiotics and demonstrate how it traverses experimental space in a way that the model achieves the same predictive performance with fewer experiments. Furthermore, through analysis of the transcriptional profiling and fitness results, we identify several cases of cross-stress protection (vulnerability), where *E. coli* treated with a biocide is more (less) protected to antibiotics than expected[52,53].

## Results

**Active learning accelerates discovery and model training.** We applied OPEX on the unexplored space of biocide and antibiotic interactions, where we performed genome-wide transcriptional profiling of *E. coli* under sequential biocide–antibiotic stress combinations (see "Methods"). We selected four antibiotics based on their diverse mechanisms of action and ten biocides based on their widespread use in hospitals and households (Fig. 2a). GPs were trained to model the genome-wide gene expression for all combinations and then used OPEX to guide 30 cycles of experimentation (i.e., the gene expression dataset in Supplementary Data 1). Each OPEX cycle resulted in a different biocide–antibiotic combination to explore (Fig. 2a), with the GP-based model being retrained with each new dataset obtained. OPEX used 44% less data to achieve the same accuracy early on (iteration 15 vs. 27, $p$ value $= 2.2 \times 10^{-16}$, Fig. 2b, Supplementary Fig. 1) and led to a better model (22% less mean average error, $p$ value $= 6.7 \times 10^{-97}$, Fig. 2c) compared to when the experiments were picked at random or by expert sampling (see "Random sampling and expert sampling" section of "Methods"). OPEX was found to be robust to noise, batch size, and dataset heterogeneity (see Section 4.1.2 of Supplementary Information, Supplementary Figs. 2–9, and Supplementary Data 2). Similar results were observed when running OPEX on all the genes of *E. coli* (Section 4.2.5 in Supplementary Information, Supplementary Figs. 10–13) or implementing OPEX by a Query-by Committee approach (Supplementary Table 2). The performance of expert sampling was always worse than that of random sampling (Supplementary Fig. 14) regardless of the level of exploration.

**Broader exploration followed by fine-tuning as a strategy.** We inspected how OPEX selects the next experiment to perform and the reason for its superior performance. We found that the distance amongst gene expression profiles for consecutively selected conditions first increases (the first ten conditions, $R^2 = 0.31$, $p$ value <0.046) and then rapidly decreases ($R^2 = 0.77$, $p$ value $= 6.7 \times 10^{-7}$, Fig. 2e). No such pattern was observed with random or expert sampling (Fig. 2d and Supplementary Figs. 15, 16). OPEX works by balancing the need to explore under-sampled experimental neighborhoods to that of picking conditions to sample for which the model is less confident, even when those are in a well-sampled area. The gain from a broader exploration and then a fine-tuning strategy is maximized when a balanced trade-off between exploration and exploitation exists (Section 4.2.3 of Supplementary Information, Supplementary Fig. 17). We investigated the behavior of OPEX without exploration as the prediction error goes up at the last few iterations. For details, see Supplementary Figs. 18–20.

**OPEX accelerates the discovery of gene expression clusters.** Next, we applied OPEX to evaluate how previous exposure to biocides would affect *E. coli*'s survival in four antibiotics with diverse modes of action. We found that biocide–antibiotic combinations are clustered in three groups based on their conditions, with two of those clusters corresponding to Rifampicin and alcohols, respectively (Fig. 3a, b, and Supplementary Fig. 21). OPEX-guided modeling was able to predict the gene expression profile and cluster membership of the unobserved conditions accurately early on (accuracy of 96.5% ± 1.9% using OPEX vs. 93.3% ± 3.3% using random sampling at iteration 15, $p$ value $= 2.4 \times 10^{-6}$), hence accelerating knowledge discovery and outperforming models that were trained based on either random or expert sampling (Fig. 3c).

**Exposure to biocides confers cross-stress protection.** Interestingly, both strong cross-stress protection and cross-stress

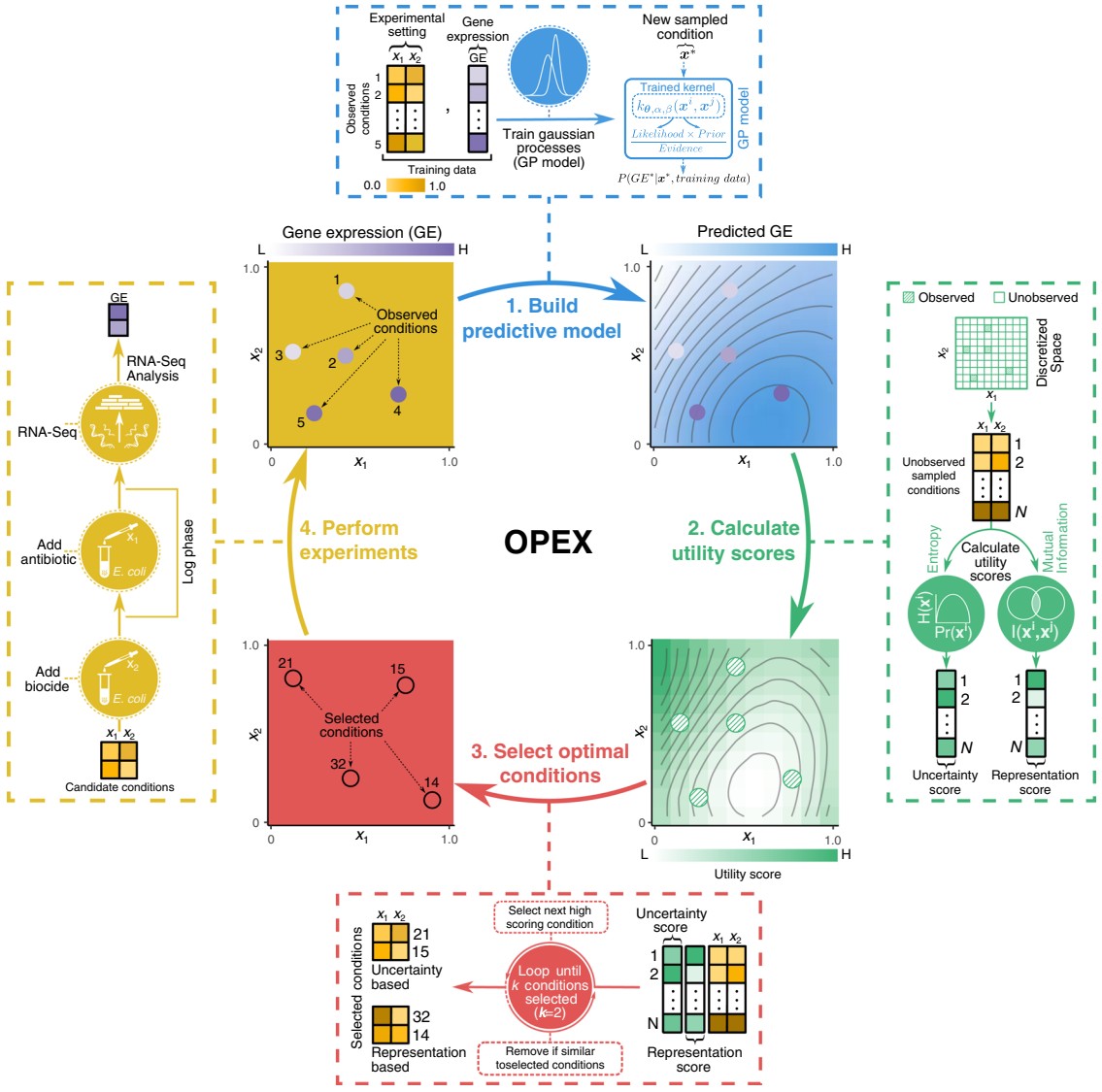

**Fig. 1 Overview of the optimal experimental design (OPEX) framework.** First, OPEX builds a predictive model of gene expression by training an underlying model (Gaussian process here) on the observed conditions and their measured gene expression values. Second, it calculates uncertainty or representation utility scores for unobserved conditions, based on entropy or mutual information metrics that are calculated from the trained GP model. Third, unobserved conditions which have the highest utility scores but are not similar to each other, are selected as candidate conditions for experimentation. Finally, new wet-lab experiments are performed to measure the cellular gene expression of candidate conditions. The newly selected conditions and their measured gene expression values are then added as an observed condition to be used for a subsequent round of OPEX. In this example, each condition is characterized by a particular choice of antibiotic and biocide depicted as $x_1$ and $x_2$ values determining a single cross-resistance experiment. The OPEX framework also allows exploring different concentrations for a biocide and antibiotic combinations (Section 2.4 in Supplementary Information).

vulnerability were observed (Fig. 4 povidone-iodine/kanamycin and norfloxacin/chlorophene). Cross-stress protection (vulnerability) is the phenomenon where exposure of an organism to a given stressor increases (*decreases*) its fitness when subsequently exposed to a different stressor[53,54]. Here, we introduce the cross-stress protection index (CSPI) to indicate any cross-stress protection or vulnerability (see "Methods"). Out of 40 cross-stress combinations, we identified cross-stress protection in 29 cases (avg. fitness increase by $38.4 \pm 10.4\%$), and cross-stress vulnerability in 4 cases (avg. fitness decrease by $2.3 \pm 0.44\%$), as shown in Fig. 4.

**Chaperons and membrane proteins are involved in cross-stress.** To establish the potential DEGs contributing to cross-stress

behavior between biocides and antibiotics, we analyzed the extreme condition pairs, with the highest and lowest fitness in our dataset. The povidone-iodine/kanamycin combination had the highest cross-stress protection ($13.0 \pm 2.1\%$ fitness increase), with only five DEGs (*htpG, dnaK, groL, groS,* and *clpB*) with a minimum two-fold increase over the control, all of which were chaperones (Fig. 5a, b). Both antimicrobials target cell proteins (Supplementary Table 1). Povidone-iodine acts non-selectively on cell proteins, resulting in oxidation, coagulation, and loss of function[54], while kanamycin, an aminoglycoside, interferes with protein synthesis by binding to ribosomes[55]. Hence, a response focused on assisting protein refolding and ensuring functional conformation would act synergistically to increase fitness in such antimicrobial combination. Our results are in accordance with previous reports that the upregulation of chaperones facilitates

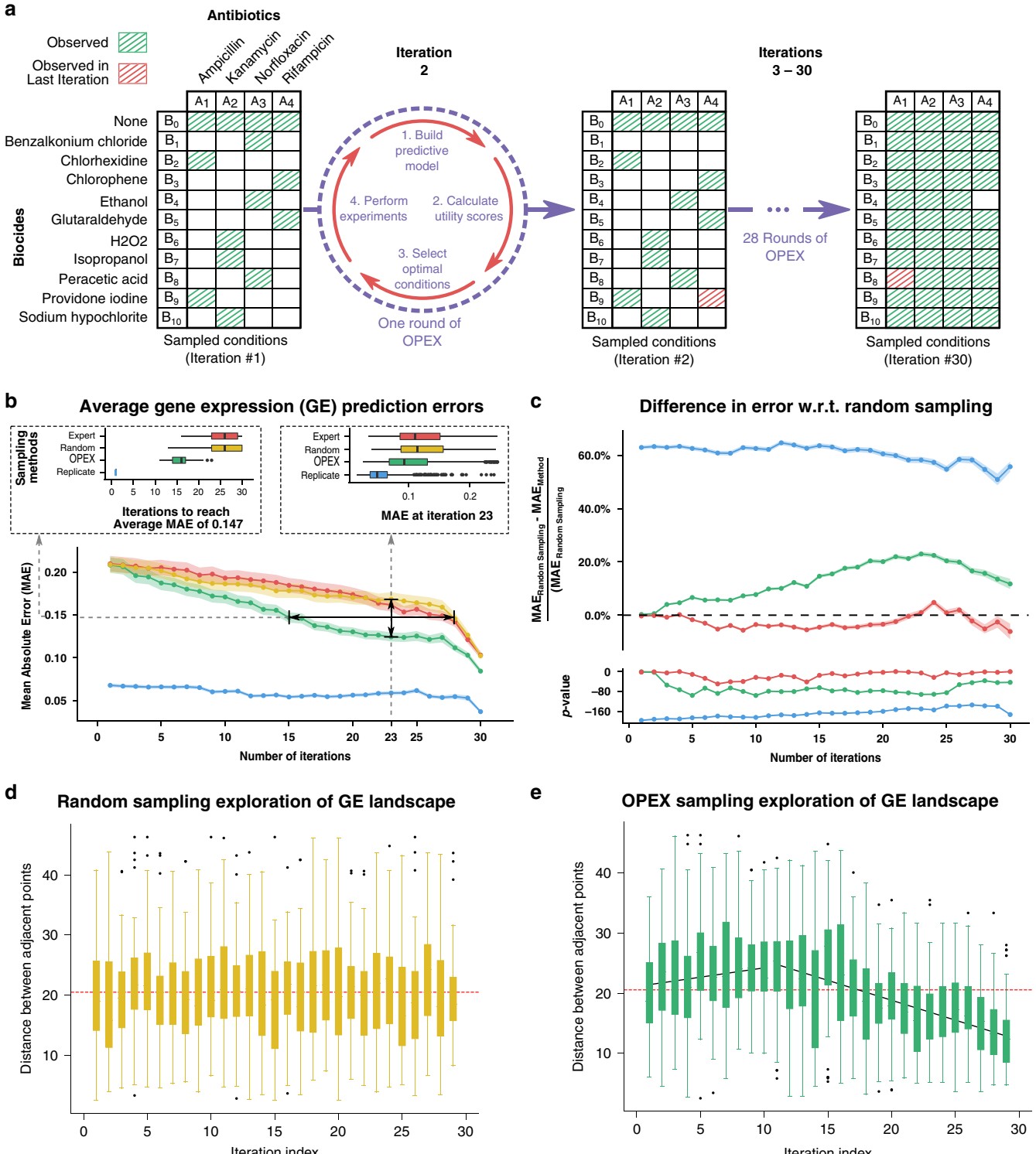

**Fig. 2 OPEX applied to biocide–antibiotic experimental space. a** After 15 randomly sampled initial conditions, we performed 30 iterations (rounds) of OPEX, where at each iteration, the genome-wide transcriptional profile of *E. coli* is measured under a biocide–antibiotic combination. **b** Prediction error (MAE) for the remainder unobserved conditions at each iteration and for the three methodologies (Expert, Random, OPEX). Background envelop depicts the 95% confidence interval after 50 random initializations. Blue line depicts the variation in expression within the experimental replicates for each condition (three biological replicates). The boxplot on the top left shows the number of iterations necessary to reach a MAE of 0.15. The boxplot on the top right shows the MAE from each sampling method at iteration 23, where the gap between MAE of random sampling and OPEX is maximum. **c** The MAE difference and *p* value for each method with respect to random sampling. The *p* values were calculated by a one-sided paired *t* test (*n* = 1123). **d, e** The distance between the gene expression profiles of every two consecutively selected culture conditions by random sampling and OPEX, respectively. Each box in the boxplot shows the average distance for 50 random initializations. The red dashed line shows the median of the distances over all the iterations. Each box represents an interquartile range, which consists of data points between the 25th and 75th percentiles. The whiskers extend to the maximum and minimum values but no further than 1.5 times the interquartile range for a given whisker.

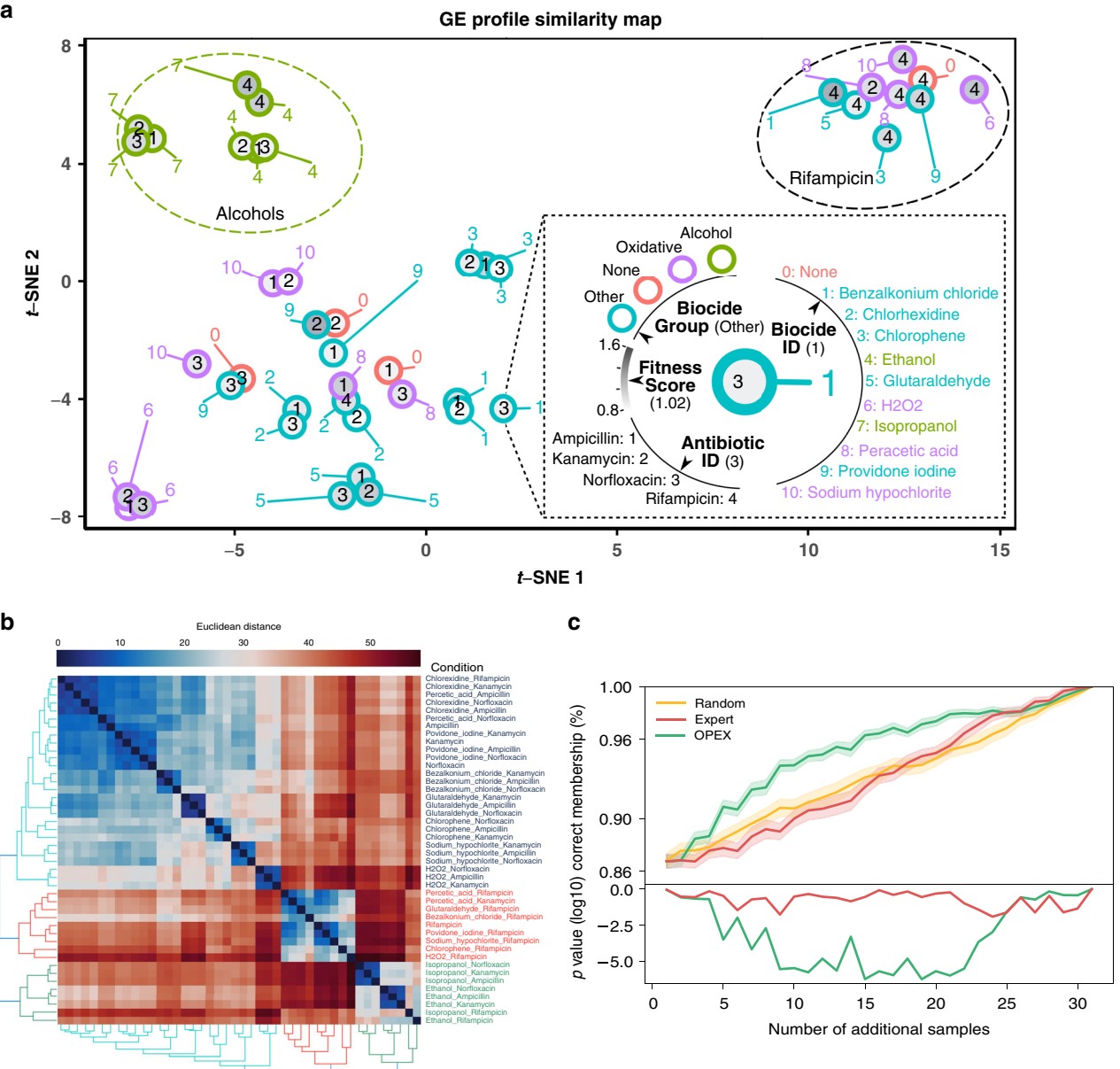

**Fig. 3 OPEX accelerated the discovery of cluster memberships for the culture conditions. a** A two-dimensional representation of the gene expression profiles of all 44 culture conditions through t-SNE, where three clusters are observed (Rifampicin, Alcohols, and Others). Each bordered circle represents a culture condition, with the border color encoding the biocide group. The grayscale color of the inner circles encodes the fitness of *E. coli* under each condition. The numbers inside and outside each circle encode for the antibiotic and biocide, respectively. **b** Hierarchical clustering of culture conditions using the transcriptomic profiles for each culture condition. **c** Performance of OPEX in predicting cluster memberships. Starting with 15 observed culture conditions, OPEX predicts more accurately than other methods the gene expression phenotype of the remaining conditions and their respective clusters (Alcohols, Rifampicin, and Others), with *p* values being calculated with reference to random sampling. The shadow belt around a curve represents plus and minus standard deviation in 50 runs. The *p* values were calculated by a one-sided *t* test.

cell survival against aminoglycosides[56]. We further validated the genetic basis of cross-stress protection by single knockouts (*p* values of 0.007–0.021, Fig. 5c). The knockout of *cpxP*, a chaperone[54], which was not differentially expressed in povidone-iodine and kanamycin combination, was used as a negative control.

Similarly, the most striking case of cross-stress vulnerability was the application of chlorophene succeeded by norfloxacin (2.7 ± 2.2% fitness decrease). We identified 12 upregulated and 9 downregulated genes exclusively in such combination (Fig. 5d, e). We focused on downregulated genes (*gstA*, *dadA*, *yhiI*, *gorA*, and *cspD*) that have been shown previously to interfere with the cellular

states, including inhibition of DNA replication, stress response, drug detoxification, and export[57–63]. Given their importance in cell protection and stress response, such downregulation could be driving the lower fitness (cross-stress vulnerability) observed compared to other antimicrobial combinations. Knockout mutant experiments validate their statistically significant role in the observed cross-stress vulnerability cases (Fig. 5f; *gstA*, *p* value = 0.0043; *dadA*, *p* value 0.0007; *yhiI*, *p* value = 0.004). The knockout of *baeS*, a gene involved in adaption to envelope stress[61], which was differentially expressed in the chlorophene and norfloxacin combination, was used as a negative control.

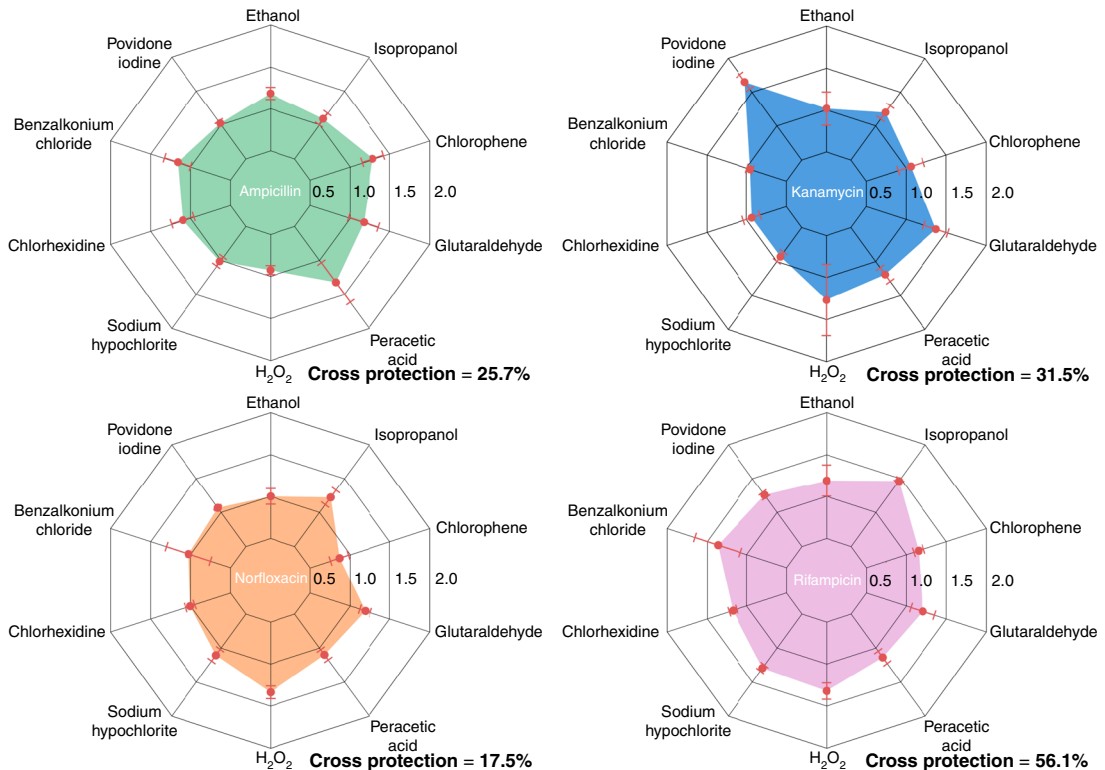

**Fig. 4 Cross-stress behavior plots of *E. coli* exposed to biocide, followed by an antibiotic.** The cross-protection percentage for each antibiotic represents the relative difference of the shaded area compared to that of the decagon with no cross-stress behavior. A cross-stress protection index (CSPI) of more than 1 denotes cross-protection of biocide-treated cells in antibiotic compared to biocide-untreated cells. CSPI less than 1 indicates cross-vulnerability of biocide-treated cells in antibiotic compared to the biocide-untreated cells. The error bars represent the standard error of mean among 3–4 biological replicates centered at their mean.

## Discussion

Our results argue that active learning and optimal experimental design can be applied to omics data collection to accelerate biological discovery significantly. We present an active learning method, OPEX, which can prioritize experimentation of unexplored neighborhoods over experimental conditions with high confidence in its predictions. When applied to unobserved combinations of antiseptics and antibiotics, OPEX led to a more accurate model faster and accelerated the process of discovering knowledge about the clustering of the 40 culture conditions. OPEX can be applied to cases where culture conditions are defined by continuous variables to encode for effects such as dosage and temporal changes, as we demonstrated with the use of synthetic data how this can be achieved with OPEX (Sections 2.5.1 and 2.5.3 in Supplementary Information). OPEX outperformed random sampling despite various levels of skewness and noise in synthetic gene expression data (Section 4.1.2 in Supplementary Information). We envision that OPEX can be generalized to other studies in which the relationship between cultural conditions and biological assays is of interest (e.g., to find the optimal culture condition for maximizing the production of specific enzymes).

The modular design of OPEX allows the integration of different machine learning models and OED sampling strategies. To that end, we used OPEX with different query-by-committee strategies together with various machine learning methods (neural networks, linear regression, and support vector regression) and achieved comparable results (Sections 2.1 and 4.2.6 in Supplementary Information). This work has demonstrated how active learning can be applied in omics experimentation and can be a precursor of predictive techniques that will guide experimentation, data processing, and discovery in life sciences at a faster pace. Considering the increase in data-driven biological research, and ease of access to corresponding data from public databases, paradigm-shifting applications of OPEX-like frameworks in biotechnology are on the horizon.

## Methods

**Culture conditions, RNA extraction, and transcriptomics.** *Escherichia coli* MG1655 was grown in minimal media (M9) with 0.4% w/v glucose for 12 h (mid to late exponential growth, optical densitiy (OD$_{600}$) ~0.8) at 37 °C. Next, 30 μL were added to 3 mL M9 0.4% glucose tubes containing one of ten biocides or no biocide. The concentrations used were sub-inhibitory. After 7–12-h growth, when cells reached mid to late exponential phase (OD$_{600}$ 0.6–1.0), the appropriate volume of one out of four antibiotics was added, and the tubes were incubated for an additional hour. Tubes without any antimicrobial (antibiotic or biocide) served as controls. To stop bacterial growth and stabilize cellular RNA, 1.5 mL of ice-cold Phenol/Ethanol (5% (v/v) phenol in ethanol) was added to per 3 mL sample. The cells were pelleted by centrifugation at 3166 g at 4 °C for 10 min and stored at −80 °C until further use. All experiments were performed in triplicate.

Total RNA was extracted from the bacterial samples using the RNeasy mini kit (Qiagen), and any possible DNA contamination was removed by performing on-column DNAse digestion (Qiagen). mRNA was enriched by removing ribosomal RNA using the capture oligonucleotide mix (MICROBExpress, Invitrogen). RNA-Seq libraries were prepared using the KAPA Stranded RNA-Seq Library Preparation Kit for Illumina platforms (Kapa Biosystems), and instructions suggested by the manufacturer were followed. Double size selection of libraries was performed using Agencourt AMPure XP magnetic beads (Beckman Coulter), where cDNA fragments ranging from 200 to 500 bp were enriched. The concentration for each sample library was determined with Qubit 2.0 fluorometer (Invitrogen). The DNA concentration of the final pooled library was determined with the Bioanalyzer DNA high sensitivity assay (Agilent, DNA Technologies Core, UC Davis). Sequencing was performed with HiSeq4000 SR50 (DNA Technologies Core, UC Davis).

**RNA-Seq data analysis.** Adapters and low-quality reads were removed from the raw reads by Trimmomatic[64] followed by alignment to the reference genome of *E. coli* MG1655 by Bowtie2[65]. Then, the bam files generated by Bowtie2 were fed into

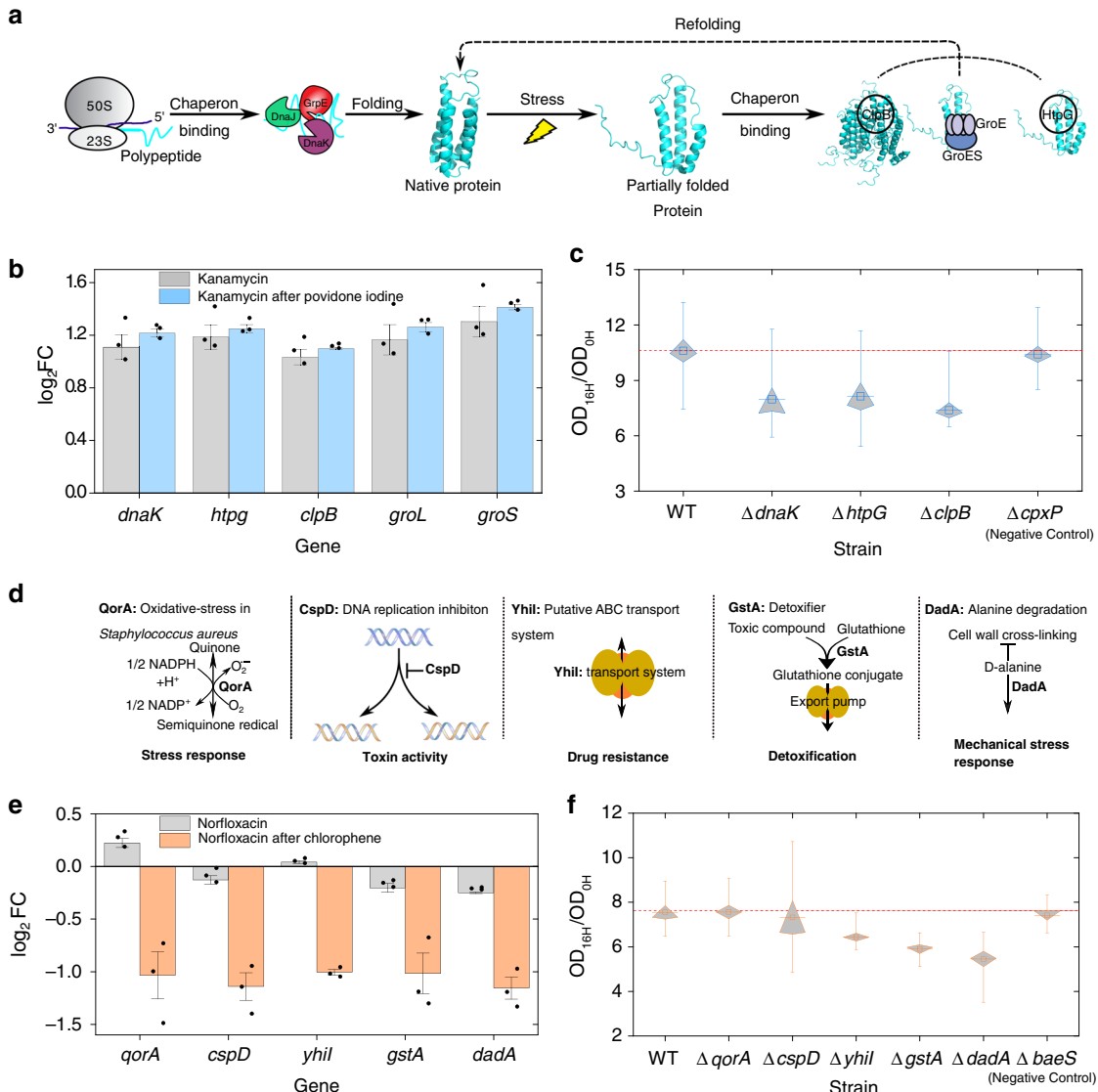

**Fig. 5 The genetic basis of cross-stress behavior. a** The protein folding pathway and key proteins corresponding to genes that were found to be differentially expressed in the povidone-iodine/kanamycin combination, which exhibits the highest cross-protection across all cases (fitness increase of 65.1 ± 10.4%). **b** All differentially upregulated genes in the povidone-iodine/kanamycin combination, compared to kanamycin only, which have all been identified as chaperons (n = 3, biological replicates). **c** Fitness change of single knockout mutants in kanamycin for each differentially expressed chaperon. The red line denotes the fitness of WT; ΔdnaK, ΔhtpG, ΔclpB are the three differentially upregulated chaperons (p value of 0.007, 0.02, and 0.001, respectively); ΔcpxP is a non-differentially expressed chaperon used as a negative control (p value of 0.396, n = 9, biological replicates). **d** Differentially downregulated stress-related genes with more than twofold change exclusively in the chlorophene/norfloxacin combination, which exhibits the highest cross-vulnerability across all cases (fitness decreases by 13.4 ± 12.9%). **e** Expression profiles of potential genes involved in cross-vulnerability. (n = 3, biological replicates). **f** Fitness change of single knockout mutants, with yhil (p value = 0.004), gstA (p value = 0.007), and dadA (p value = 0.007) knockout mutants having a lower fitness than WT after norfloxacin exposure, supporting the cross-vulnerability due to chlorophene conditioning leading to downregulation. ΔbaeS (p value of 0.479) is a differentially expressed gene, which is involved in adaption to envelope stress, was used as a negative control (n = 8, biological replicates). Black circles in **b** and **e** represent raw data points. Boxes in **c** and **f** represent the standard error of the mean (SEM), the middle line represents the mean value, and the whisker line extends to the minimum and maximum values. An error bar represents an SEM. The p values were calculated by the one-sided Wilcoxon rank-sum test.

FeatureCounts[66], yielding the number of transcripts for each gene in any given biological replicate. The number of transcripts for each gene was then converted to count per million (CPM) after dividing by the library size of the replicate. Finally, the gene CPM values for each replicate were normalized using the means of the corresponding gene CPM values from the control sample replicates. In all figures related to the performance of OPEX except Supplementary Figs. 10–13, only the genes that have CPM larger than 100 in half of the replicates were used[67].

**Modeling gene expression using GPs**. We modeled the expression level of a gene under an experimental condition by a GP, considering that gene expression levels under similar conditions are similar. When predicting expression level under a new experimental condition, a GP model generates a probability density function. The function is then used by OPEX, to calculate a utility score to estimate the utility of performing wet-lab experiments for this new condition to improve model predictions. For mathematical details, see Section 1.2.1 in Supplementary Information. The mlegp package is used to train the GP presented here[68]. We also tested other data-driven models for gene expression prediction including feed-forward neural network, linear regression, and support vector regression (Section 2.1 in Supplementary Information).

**OPEX algorithm**. OPEX was initialized with 15 randomly selected conditions and their corresponding gene expression profiles. We ensured to include each antibiotic

and biocide at least once in the initial dataset. In each iteration, OPEX trains a predictive model for predicting gene expression levels for all the remaining conditions. Next, utility scores (based on entropy or mutual information) are calculated for each condition using the predicted distribution of gene expression levels. OPEX can switch between exploration (conditions are randomly selected) and exploitation (conditions are selected by using a predictive model). In this work, we use GP as the predictive model, and consecutive switching between exploration and exploitation strategies in OPEX, with different switching frequencies also explored (Section 4.2.3 of Supplementary Information). We also tested other OED approaches by replacing the predictive model and the utility function of OPEX with alternative choices (Sections 2.1 and 4.2.5 in Supplementary Information).

**Random sampling and expert sampling.** For comparison with the OPEX strategy, we considered a random sampling strategy and three different expert sampling strategies by consulting independent chemists and biologists. Random sampling and the top performing expert sampling strategy are used as the baseline for evaluating the performance of OPEX. The utility functions used by the three expert sampling strategies are: (a) the pairwise structural similarity among the ten biocides and four antibiotics, (b) the similarity in the biological mechanism of the biocides and antibiotics, and (c) the relative dominance among the antibiotics and biocides. The expert sampling strategy (a) had the best performance (Supplementary Fig. 14) and was used as a benchmark in Fig. 2. These strategies follow the same workflow as described above for OPEX except that the utility function is either random sampling function or informed by expert knowledge and not by the GP model. In the random sampling strategy, the following culture condition is selected randomly from unobserved conditions. In expert sampling, the devised utility function calculates condition similarities based on their molecular fingerprints, biological mode of action, or the expected level of impact from the antibiotic on cellular transcription. After condition similarities are calculated, the condition that is most dissimilar to observed conditions is selected for the next iteration. For details, see Sections 2.2 and 2.3 in Supplementary Information.

**Fitness measurements.** Fitness experiments were performed with *E. coli* MG1655 in sublethal concentrations of all four antibiotics, pretreated with and without any of the ten biocides, in three replicates. Briefly, fresh colonies of MG1655 were transferred to 1-ml LB broth and grown overnight at 37 °C in an incubator shaker. The subsequent day, 5-μl culture was transferred to 2 ml of 0.4% glucose M9 minimal medium and grown for 12 h at 37 °C in an incubator shaker. After 12 h, 5 μl culture was transferred to 0.4% glucose M9 supplemented with or without biocide and then were grown overnight at 37 °C. The next day, 5 μl of grown culture was transferred to 195 μl of 0.4% glucose M9 medium supplemented with or without antibiotic, in a 96 well plate. Growth profiles were measured at 37 °C in a plate reader (BioTek HTX) at 600 nm. At 16 h, ODs reached stationary phase. The OD values at 0 and 16 h were used to calculate the CSPI.

**Cross-stress protection index (CSPI).** We introduce the CSPI as the normalized ratio of fitness (measured by OD at 16 h) between the treated and untreated cells to capture the effect of cross-stress protection:

$$\mathrm{Cross-stress\ protection\ index} = \frac{(\mathrm{OD\ biocide\ treated}_{16h})}{(\mathrm{OD\ biocide\ treated}_{0h})} \div \frac{(\mathrm{OD\ biocide\ untreated}_{16h})}{(\mathrm{OD\ biocide\ untreated}_{0h})}.$$

A CSPI larger than 1 corresponds to cases where the fitness, measured by growth curves, of biocide-exposed cells is higher in the presence of the antibiotic to cells that have not been exposed to the biocide (i.e., a case of cross-stress protection). In a similar fashion, a CSPI of <1 corresponds to a cross-stress vulnerability.

**Statistical validation.** To evaluate the hypothesis that GP model achieves average MAE of 0.147 for GE predictions earlier when using OPEX compared to random sampling (as illustrated in Fig. 2b top left), we used a one-sided paired *t* test with 50 degrees of freedom corresponding to 50 different random initial conditions. To evaluate the hypothesis that MAE of predicted GE for a given gene across remaining conditions is lower for OPEX compared to when random sampling is employed (as illustrated in Fig. 2c), we used a one-sided paired *t* test where the degrees of freedom is equal to the number of genes. To evaluate the significance of trends observed in Fig. 2e (increase in exploration of GE landscape until iteration 10 followed by decrease), we used the F-test value corresponding to the fitted linear models. When evaluating the hypothesis that predicted cluster memberships for remaining conditions are more accurate when OPEX is used compared to random sampling (Fig. 3c), we used a one-sided paired *t* test with 50 degrees of freedom corresponding to 50 different random initial conditions. To evaluate the hypothesis that relative fitness is lower for the knockout strain compared to wild-type (Fig. 5c, f), we used a one-sided Wilcoxon rank-sum test with nine samples each.

## Data availability
The RNA-Seq data are available in Gene Expression Omnibus under the ID GSE144604.

## Code availability
The R code for OPEX is available at https://github.com/IBPA/OPEX.

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

## Acknowledgements

We want to thank expert participants and various labs in the UC Davis Genome Center for help in identifying expert or commonly used methodologies for experiment selection. This work was supported by NSF ABI award 1743101, NSF OCI award 1516695, and XSEDE award TG-BCS180014 for computational resources, and an unrestricted gift from Mars Inc. to I.T.

## Author contributions

The OPEX algorithm was designed by X.W., I.T., and A.E. X.W. performed all computational analysis. B.M.P.P. and N.R. performed the wet-lab validation and all experiments. X.W., B.M.P.P., N.R., A.E. and I.T. wrote the manuscript. I.T. conceived and supervised all aspects of the project.

## Competing interests

The authors declare no competing interests.
