## [Peer Review File · Nature Communications]

Reviewers' Comments:

Reviewer #1:

Remarks to the Author:

This work presents an optimal experimental design method, OPEX, based on the use of Gaussian processes to automate omics data collection. As an application, the authors consider a case related to biocide and antibiotic cross-resistance in *E. coli*.

One of the major conclusions of the work, as stated in both the title and the discussion, is that optimal experimental design can accelerate knowledge discovery. Although the data presented, for the particular case of study, are consistent with this general conclusion, the result per se is not original. Several previous works have addressed the use of OED to improve models predictive capabilities and thus knowledge generation while reducing experimental effort. The authors mention some of these works but ignore many others: Balsa-Canto et al. *IET Syst. Biol.* 2008; Kreutz and Timmer, *FEBS J.* 2009; Apgar et al. *Mol Biosyst.* 2010; Sverchkov and Craven, *PLOS Comp. Biol.* 2017; Bandiera et al. *Processes*, 2018; to name a few in the context of systems and synthetic biology.

The second major conclusion is that OPEX approach offers the means to automate data collection for training Gaussian processes (or other machine learning models). In this respect, the novelty of the work may lie on the utility criteria (mostly entropy and mutual information) used in the OPEX method. In this case, it is recommended to submit a revised version of the manuscript to a more specific journal.

Some recommendations for a revised version of the manuscript include:

1. Title and abstract: The title is too broad to effectively show what the manuscript is about and should be more specific on the type of methods used or the type of knowledge gained. Same applies to the abstract. For example, neither the title nor the abstract mention Gaussian processes or Machine learning, which are a fundamental part of the development.
2. Context of the proposed strategy: The introduction should emphasise the role of the Machine learning approach with an overview of active learning or optimal experimental design in the context of Gaussian processes or alternatives such as artificial network models. The authors do not mention any other previous works. Again the work by Sverchkov and Craven, and others cite therein seem pertinent.
3. Reproducibility and future use of the OPEX approach: The authors should provide the means to reproduce their results, i.e. detailed description of methods and software tools used or the scripts/codes to reproduce the analyses but also a fair comparison with other approaches used by the Machine learning community.

It is also suggested to discuss the following points:

1. How general is the OPEX approach? The case considered in the manuscript regards a scenario with a finite number of samples / experimental conditions. Would it be possible that OPEX automatically selects biocide/antibiotic doses (from a continuous space of solutions)? In addition, the analysis of gene expression from time series underpins many biological studies. Would the approach scale /generalise to these more complex scenarios?
2. The example considers the design of different cultures (biocides and antibiotics combinations). It seems that the selection is made from a pre-defined set of 45 conditions. How were those conditions selected? Also, it would be nice to use a different set of experiments for validation (never taken for training) to compare all iterations and test for model extrapolation capabilities. Figures S9 and 2.B in the main text show that the initial mean absolute error is around 0.22, which is indeed quite close to 0.147 used as a reference for the success of OPEX. What would the results be in relative terms?
3. In general, changing culture conditions may imply substantial differences in gene expression;

inactive genes in one condition may be active in others. Mistaken removal can compromise any further analysis. It would be nice to update the OPEX method to deal with this.

4. How are the hyper-parameters of the Gaussian process computed? Was global convergence assured? This relates to the possibility of reproducing results.

5. How are cross-protection and cross-vulnerability defined?

Reviewer #2:

Remarks to the Author:

The main finding of this study is that the application of an Optimal Experimental Design Framework was successful in selecting the most informative experiments to perform in order to maximize information output. Thereby selecting experiments favoring broader exploration followed by fine-tuning emerged as the optimal strategy. It was demonstrated that 44% less data were needed to achieve the same accuracy early on. In Figure 4D the authors show cross-stress behavior plots of *E. coli* exposed to biocide, followed by an antiseptic. How many transcriptional profiles were included in this analysis? All of them? I would have liked to see the results with only 15 transcriptional profiles included and then how the results of the OPEX compares to expert and random. In other words how many experiments would you need in OPEX, Expert or random to identify e.g. the cross protection of Povidone iodine and Kanamycin? I would have liked to see this in order to judge whether the early on more accurate information via OPEX is enough and useful to make predictions that can be experimentally verified.

Reviewer #3:

Remarks to the Author:

This paper presents an optimal experimental design approach, called OPEX, that accelerates knowledge discovery to understand transcriptional interplay between biocides and antibiotics treatments based on omics data.

This work is timely since active learning approaches are getting increasingly popular while their true potential for learning remains to be clarified.

Regarding the methodological novelty, all the steps in the active learning loop appear relatively standard but their concatenation in the proposed manner is original and results in a clearly novel framework.

Using one case study, the authors demonstrate that indeed the proposed framework allows them to learn faster than when experiments are chosen randomly or based on expert knowledge. The learning task is relatively modest (choosing the best order of experiments to perform within 45 predefined experiments) but also not fully trivial and leading to interesting biological observations.

The quality of the illustration is very high and most of the results are nicely graphically illustrated. The main text is also well written even if some information is missing (eg, what is the expert approach used, what is the exploration strategy employed in Fig 2B) and some claims appear occasionally too strong (eg on the generalizability of the proposed approach to other omics data or predictive model classes).

The supplementary material is not well connected to the main text. A significant part is dedicated to an in silico evaluation of the approach. This part has not very clear conclusions and none of them are used in the main text. The quality of the text is also lower, with a number of small typos/errors. However, these shortcomings can be easily corrected.

Minor comments:

- in the abstract, the term of 'cross-behavior potential' is not clear.
- At several places, make a better distinction between what is due to the learning approach itself (eg faster learning) and what is due to the generation and analysis of the whole data set (eg the majority of the biological findings). For example, in the abstract, 'this led to the discovery...' incorrectly connects the proposed OED approach ('this') with biological findings (that actually result from the analysis of the full data set).
- title of Figure 3 'scientific discovery through active learning' is misleading: the findings are mainly from the entire dataset. Only panel C refers to the learning approach.

Responses to reviewers' questions

Reviewer #1 (Remarks to the Author):

One of the major conclusions of the work, as stated in both the title and the discussion, is that optimal experimental design can accelerate knowledge discovery. Although the data presented, for the particular case of study, are consistent with this general conclusion, the result per se is not original. Several previous works have addressed the use of OED to improve models predictive capabilities and thus knowledge generation while reducing experimental effort. The authors mention some of these works but ignore many others: Balsa-Canto et al. IET Syst. Biol. 2008; Kreutz and Timmer, FEBS J. 2009; Apgar et al. Mol Biosyst. 2010; Sverchkov and Craven, PLOS Comp. Biol, 2017; Bandiera et al. Processes, 2018; to name a few in the context of systems and synthetic biology.

The second major conclusion is that OPEX approach offers the means to automate data collection for training Gaussian processes (or other machine learning models). In this respect, the novelty of the work may lie on the utility criteria (mostly entropy and mutual information) used in the OPEX method. In this case, it is recommended to submit a revised version of the manuscript to a more specific journal.

Response: We agree with the reviewer that OED is not a novel framework in biology, and as the reviewer also notes, we have mentioned this in our manuscript (“In biology, OED...drug discovery”, lines 55-56). We are not stating that OED is a novel method in biology, and it is to be expected as OED/active learning is a field of research that is both extensive in scope and active since the 1950's. We would like to note, however, that OED has never been used for selecting omics experiments. For example, the papers that the reviewer cites are focused on parameter selection with other formulations and other types of data: in *Balsa-Canto et al. IET Syst. Biol. 2008* the authors studied formulating optimal experimental design as a dynamic optimization problem to calibrate the parameters of cell-signaling cascade models, similar to *Apgar et al. Mol Biosyst. 2010*, where the authors focused on imputing parameters at a biochemical and pharmacokinetic level. Similarly, in *Bandiera et al. Processes, 2018*, the authors used OED for parameter calibration in synthetic biology. We have now added the references to those referenced already on the manuscript (ref [44-48], line 40).

As such, the novelty of this paper resides in (a) applying OED in omics experiments, especially in the case of transcriptional profiling, (b) its novel OED problem formulation that is tailored for omics data, (c) demonstrating that OED leads to more accurate gene expression models in a shorter time than other methods (lines 87-92) and that it accelerates biological discovery (the clusters of the 40 culture conditions, lines 106-113) that we validated by performing forward validation in an unexplored biological setting (biocide/antibiotic exposure), (d) providing clear biological insights regarding the potential and genetic basis of cross-stress behavior (lines 115-150). We also highlighted the point that there is additional work to be done for the application of OED methods in biology (lines 164-167 & 175-177).

Some recommendations for a revised version of the manuscript include:

1. Title and abstract: The title is too broad to effectively show what the manuscript is about and should be more specific on the type of methods used or the type of knowledge gained. Same applies to the abstract. For example, neither the title nor the abstract mention Gaussian processes or Machine learning, which are a fundamental part of the development.

Response: We have now revised much of the abstract and introduction to clarify the methods used. We believe that it is better to not discuss about gaussian processes or other details (e.g. using entropy and mutual information to encode the utility function) in the abstract, as OPEX is a framework that can work

with multiple models. Specifically, since the models that we investigated include SVMs, ANNs, Linear Regression, in addition to Gaussian processes (GPs), we don't want to emphasize the application of a specific modeling method (GP here), rather, the potential of OED as a methodological framework to accelerate knowledge discovery within the context of omics experimentation. In this context, the specific OED and model implementation is a proof-of-concept instance of the many possible and illustrates how this concept can materialize. To make this point clear, we have added this description in the Discussion section (lines 169-177), Methods section (lines 231-233), and Supplementary Material (lines 190-208, and 461-483).

2. Context of the proposed strategy: The introduction should emphasise the role of the Machine learning approach with an overview of active learning or optimal experimental design in the context of Gaussian processes or alternatives such as artificial network models. The authors do not mention any other previous works. Again the work by Sverchkov and Craven, and others cite therein seem pertinent.

Response: We have now expanded the introduction section to explain the Gaussian processes method and its applicability for OED in the context of omics (lines: 61-67). We also discuss other alternative choices in the discussion section (lines 169-172). We now reference Sverchkov and Craven in the Introduction section (ref: 47). We list a few publications about artificial network models in the context of active learning [3-4].

3. Reproducibility and future use of the OPEX approach: The authors should provide the means to reproduce their results, i.e. detailed description of methods and software tools used or the scripts/codes to reproduce the analyses but also a fair comparison with other approaches used by the Machine learning community.

Response: Our code along and its documentation is now available on Github at the following address: <https://github.com/IBPA/OPEX>. We also added this information in the manuscript (lines 299-301).

To compare with other methods, we surveyed OED methods and then adapted three query-by-committee methods for comparison. Specifically, we used query-by-committee using different types of models, query-by-committee using bootstrapping, and D-optimal experimental design. We evaluated these three OED approaches using the same framework of OPEX. All approaches differ in the utility function used while the last approach also employs a different predictive model. The performance of OPEX and three OED approaches are summarized in Table S1 of Supplementary Materials. The performance varied from significant to no advantage when compared to random sampling, and in all cases similar or less advantage to that observed when OPEX is used. With respect to the overall improvement of MAE relative to random sampling in all iterations, OPEX with entropy achieved 12.7% while the best non-OPEX method (query by committee using different types of models) achieved 11.0% indicating a slight advantage for OPEX with entropy (p-value = 9×10^{-9}). See Supplementary Material, section 4.2.6 and Fig. S19 for further details.

It is also suggested to discuss the following points:

1. How general is the OPEX approach? The case considered in the manuscript regards a scenario with a finite number of samples / experimental conditions. Would it be possible that OPEX automatically selects biocide/antibiotic doses (from a continuous space of solutions)? In addition, the analysis of gene expression from time series underpins many biological studies. Would the approach scale /generalise to these more complex scenarios?

Response: We agree with the reviewer that OPEX can be generalized to indicate dosage and other more complex scenarios. We provide a synthetic data example where we show that this is possible

(Supplementary Material, lines 260-263 and 335-369). In general, our results show that OPEX performs better than random sampling, given datasets with initial sizes, noise levels, skewness as well as various exploration frequencies (See Fig. S2-S8). This point is also discussed in the Discussion section of the manuscript (lines 159-167). Similarly, OPEX can be used for time series data, something that we now also mention in the Discussion section (lines 159-167).

2. The example considers the design of different cultures (biocides and antibiotics combinations). It seems that the selection is made from a pre-defined set of 45 conditions. How were those conditions selected? Also, it would be nice to use a different set of experiments for validation (never taken for training) to compare all iterations and test for model extrapolation capabilities. Figures S9 and 2.B in the main text show that the initial mean absolute error is around 0.22, which is indeed quite close to 0.147 used as a reference for the success of OPEX. What would the results be in relative terms?

Response: We selected four antibiotics that represent different mechanisms of action (ampicillin, inhibition of cell wall synthesis; kanamycin, inhibition of translation; Norfloxacin, inhibition of replication; and Rifampicin, inhibition of transcription) and ten biocides that are the most widely used in the hospitals and households. We have added this clarification in the main manuscript (lines 82-84) and in Supplementary Materials on the selection method and pertinent data (lines 315-317).

We would like to note that all 45 conditions (10 biocides, 4 antibiotics, 1 control, all in 3 biological replicates, so 135 RNA-Seq samples) are new experiments that we conducted, and for all the combinations, genome-wide transcriptional profiling on those conditions has never been performed before. When we train OPEX, we train it on 15 of those conditions, and use the rest 30, as unexplored conditions – the “validation set” - with OPEX having no access to any of that data. Then OPEX chooses the next condition to explore, we “explore” it by providing the RNA-Seq data to OPEX, which in turn retrains the models and selected the next condition, and so on. This is identical to the scenario that the reviewer is proposing in terms of validation, with 15 conditions as training set, and 30 conditions as the validation set (and since this is active and not supervised learning, the validation condition appear one by one). The fact that we did the RNA-Seq experiments *a priori* and mapped the whole space, it was careful planning and efficient experimental design to use multiplexing in the sequencing lanes, and has nothing to do with actual training and testing/validation of the method (i.e. there is no “pollution” of the training set from the validation data, at any step of the experimental or computational design).

Furthermore, to ensure statistical validity and robustness of the results, we re-ran OPEX 50 times, each time with a different random seed of 15 conditions (out of the 45). In each run, OPEX parameters were randomly initialized, and there was no “memory” or information flow among any of the runs. All claims and statistics are based on all 50 runs. We want to emphasize that if one was to run just a single experiment (15 training and 30 “validation” conditions), which is equivalent to what we did but without the randomization, the results could be overfitting to that specific selection of the 15 training conditions. Finally, on the synthetic data analysis, the performance was also measured on a hold-out dataset (See Supplementary Material, lines 274-264).

In terms of relative comparison of the error, we agree that this is a useful comparison, and lines 85-92 of the manuscript summarize the findings in percentages: “Each OPEX cycle resulted in a different biocide-antibiotic combination to explore (Fig. 2A), with the Gaussian process-based model being retrained with each new dataset obtained. OPEX used 44% less data to achieve the same accuracy early on (iteration 15 vs. 27, p-value = 2.2×10^{-16} , Fig. 2B) and led to a better model (22% less mean average error, p-value = 6.7×10^{-97} , Fig. 2C)”. Also note that the standard deviation of gene expression among the replicates given the same culture condition is around 0.06 (see blue line in Fig. 2B), which can be used to better interpret the significance of mean absolute error differences.

3. *In general, changing culture conditions may imply substantial differences in gene expression; inactive genes in one condition may be active in others. Mistaken removal can compromise any further analysis. It would be nice to update the OPEX method to deal with this.*

Response: To address this point, we ran OPEX based on all 4,391 genes (Supplementary Material, lines 463-479), instead of only the 1,123 genes that have a CPM score more than 100. The performance is similar, with OPEX reaching similar advantage in MAE and with a 44% reduction in data points (Fig. S20 and Table S1). We clarify this point also in the manuscript (lines 91-92).

4. *How are the hyper-parameters of the Gaussian process computed? Was global convergence assured? This relates to the possibility of reproducing results.*

Response: The hyper-parameters (the parameters of the kernel function) are trained using maximum likelihood estimator through Broyden–Fletcher–Goldfarb–Shanno (BFGS) algorithm [1], which is an iterative quasi-Newton method used by the MLEGP package that was used in this work [2]. We can reproduce the results of this study as we have introduced and saved the seed point for the random number generator engine. The information on how to reproduce the results is in the Github information page of OPEX (<https://github.com/IBPA/OPEX>). Global convergence is not assured since the problem is non-convex.

5. *How are cross-protection and cross-vulnerability defined?*

Response: To avoid confusion, we have changed the wording to the terms “cross-stress protection” and “cross-stress vulnerability” throughout the manuscript. *Cross-stress protection* is the phenomenon where exposure of an organism to a given stressor increases its fitness when subsequently exposed to a different stressor (references 50-51; lines: 117-120). Similarly, *cross-stress vulnerability* occurs when the exposure to a stressor makes the cells less fit (measured by any fitness assay, such as growth curves, survival assays, competition assays, among others) when subsequently exposed to another stressor. We have now introduced the term “cross-stress protection index (CSPI, lines: 260-268)” as the normalized ratio of fitness (measured by OD measurements at 16h) between the treated and untreated cells to capture the effect of cross-stress protection:

$$\text{Cross-stress protection index} = \frac{(OD \text{ biocide treated}_{16h})}{(OD \text{ biocide treated}_{0h})} \div \frac{(OD \text{ biocide untreated}_{16h})}{(OD \text{ biocide untreated}_{0h})}$$

A CSPI larger than 1 corresponds to cases where the fitness, measured here by growth curves, of biocide-exposed cells is higher in the presence of the antibiotic to that of cells that have not exposed to the biocide (i.e. a case of cross-stress protection). In a similar fashion, a CSPI of less than 1 corresponds to a cross-stress vulnerability. In Fig. 4, Biocide-Antibiotic combinations with CSPI >1 exhibit cross-stress protection and <1 cross-stress vulnerability.

We would like to thank the first reviewer for their time and helpful comments.

Reviewer #2 (Remarks to the Author):

The main finding of this study is that the application of an Optimal Experimental Design Framework was successful in selecting the most informative experiments to perform in order to maximize information output. Thereby selecting experiments favoring broader exploration followed by fine-tuning emerged as the optimal strategy. It was demonstrated that 44% less data were needed to achieve the same accuracy early on. In Figure 4D the authors show cross-stress behavior plots of E. coli exposed to biocide, followed by an antiseptic. How many transcriptional profiles were included in this analysis? All of them?

Response: Nine transcriptional profiles were used for Fig. 4D (what is now Fig. 5D), but we think that the reviewer is referring to the cross-stress behavior plots of Fig. 3D (what is now Fig. 4). For that figure, no transcriptional profiles have been used, only growth curves. We understand from the reviewer's question that having the OPEX biological discovery (Fig. 3A-C) together with the analysis of the cross-stress behavior (what used to be Fig. 3D), can be misleading. For Fig. 3A-C the focus is on OPEX and how it performs on predicting cluster membership based on expression, while on Fig. 3D the focus was not on OPEX rather the biological insights themselves. For that reason, we moved the cross-stress behavior plots (previous Fig. 3D) in a separate figure (Fig. 4) to avoid confusion.

I would have liked to see the results with only 15 transcriptional profiles included and then how the results of the OPEX compares to expert and random. In other words how many experiments would you need in OPEX, Expert or random to identify e.g. the cross protection of Povidone iodine and Kanamycin? I would have liked to see this in order to judge whether the early on more accurate information via OPEX is enough and useful to make predictions that can be experimentally verified.

Response: To the reviewer's point regarding "results with only 15 transcriptional profiles", we would like to clarify that indeed OPEX uses only 15 conditions (and since each condition has three biological replicates, we have 45 RNA-Seq samples for the 15 conditions) to predict the next condition. To be more specific, initially OPEX is trained on 15 conditions (a parameter in our setting that can be changed; in general, this is the conditions/experimental settings which we have already explored), and uses the remaining 30, as unexplored conditions – the "validation set" - with OPEX having no access to, or taking into account in any way. Then OPEX chooses the next condition to explore (one of the 30). We "explore" it by providing the RNA-Seq data to OPEX, which in turn re-trains the models and selects the next condition (of the remaining 29), and so on. This is identical to the scenario in which we have 15 conditions as the training set, and 30 conditions as the validation set (and since this is active and not supervised learning, the validation condition appears one by one). The fact that we performed the RNA-Seq experiments *a priori* and mapped the whole space, is because we need to be efficient in using multiplexing in the sequencing lanes, and has nothing to do with actual training and testing/validation of the method (i.e. there is no "pollution" of the training set from the validation data, at any step of the experimental or computational design). In addition, to ensure statistical validity and robustness of the results, we re-ran OPEX 50 times from scratch (i.e. each run is independent, no parameters or any other memory from one run to another), each time with a different random seed of 15 conditions (out of the 45 conditions total), with statistics being based on all 50 runs. For more information regarding this point, please see point 2 from Reviewer #1, as well as Supplementary Materials (lines 284-296).

Regarding accelerated knowledge discovery, the claim we make is related to gene expression and clustering of conditions. More specifically, we show that OPEX leads to more predictive models of gene expression in less time or data points (lines 85-92, Figure 2 B-C), which in turn be used to accurately cluster similar conditions in terms of overall gene expression, even before those conditions have been explored as we demonstrated later on (lines 106-113, Figure 3C). We do not claim that OPEX can discover cross-stress protection behavior in different conditions (in terms of DEGs or fitness scores), as that trait is measured with growth curves, and not based on gene expression – which is what OPEX is focused on and optimizes. Cross-stress protection may be indirectly related to gene expression, and it can potentially be predicted by having differentially expressed genes as features, however, without any other information (e.g. which genes are the most informative/relevant), such predictor will need a large training set (thousands of samples) even with regularization schemes, due to the large number of features (genes, among others). Additionally, creating a scenario where OPEX is applied to predict cross-stress protection directly might be possible if contextual information (e.g. the mechanism of action of each stressor) is provided as an input. Given the specificity of this trait on each pair combination, we again anticipate the need for a large dataset and still its performance would be uncertain, so this investigation is outside the scope of this work.

We would like to thank the second reviewer for their time and helpful comments.

Reviewer #3 (Remarks to the Author):

The quality of the illustration is very high and most of the results are nicely graphically illustrated. The main text is also well written even if some information is missing (eg, what is the expert approach used, what is the exploration strategy employed in Fig. 2B) and some claims appear occasionally too strong (eg on the generalizability of the proposed approach to other omics data or predictive model classes).

Response: Thank you for the comments. There are 3 expert approaches, that are described in Supplementary Methods (lines 209-238) and we have now added a summary in the Methods section of the manuscript (lines 235-249). Briefly, the three approaches are based on one of these three metrics (a) the pairwise structural similarity among the 10 biocides and 4 antibiotics, (b) the similarity between the mechanism of action for the various biocides and antibiotics, (c) the relative dominance among the antibiotics and biocides. In the main manuscript, we tested all three, and we used the best performing of the three (pairwise structural similarity) for comparison to OPEX. We have also clarified the exploration strategy in the revised description of the OPEX algorithm (lines 227-233). More specifically, in each iteration, OPEX, trains a predictive model for predicting gene expression levels for all the remaining conditions. Next, utility scores (based on entropy or mutual information) are calculated for each condition using the predicted distribution of gene expression levels. OPEX can switch between exploration (conditions are randomly selected) and exploitation (conditions are selected by using a predictive model). In this work, we use GP as the predictive model, and we use consecutive switching between exploration and exploitation strategies in OPEX, with different switching frequencies also explored (see section 4.2.3 of Supplementary Materials). We have also revised statements that may be misleading (including the statement about the generality of the model (lines 169-177)).

The supplementary material is not well connected to the main text. A significant part is dedicated to an in silico evaluation of the approach. This part has not very clear conclusions and none of them are used in the main text. The quality of the text is also lower, with a number of small typos/errors. However, these shortcomings can be easily corrected.

Response: We have now revised the structure and re-written most of the supplementary material for clarity. This includes the organization of the supplementary material to the OPEX framework (Chapter 1), Computational Methods (Chapter 2), Experimental Methods (Chapter 3), and Results (Chapter 4). We have added a pseudocode for the OPEX algorithm and defined OPEX as a general optimization problem (equation 13) and for the specific instance that we are presenting in the paper (equation 14). The robustness of OPEX with respect to noise, batch size, and dataset heterogeneity has been evaluated with synthetic data (Supplementary Material, Section 2.5.1) and these are now referenced in the main text (lines 159-167), although their discussion is still in the Supplementary Material, Section 4.1.1.

Minor comments:

- in the abstract, the term 'cross-behavior potential' is not clear.

Response: We have changed the term to cross-stress behavior, and we define it as both 'cross-stress protection and vulnerability' (lines 117-120).

- At several places, make a better distinction between what is due to the learning approach itself (eg faster learning) and what is due to the generation and analysis of the whole data set (eg the majority of the biological findings). For example, in the abstract, 'this led to the discovery...' incorrectly connects the

proposed OED approach ('this') with biological findings (that actually result from the analysis of the full data set).

Response: Thank you for pointing this out, it can be misleading. We have corrected that sentence to avoid any confusion by clearly separating the results about OPEX and the results from the analysis of the full data set. Specially, we corrected the sentence “this led to the discovery...” in the abstract (line 9). We separated the panel D in the previous Fig. 3 as a separate Figure (current Fig. 4), as that depicted biological findings resulting from the analysis of growth curves and not from OPEX, while panels A-C from Fig. 3 are related to the demonstration of how OPEX accelerates the gene expression cluster discovery for the 40 culture conditions (clustering results coming from OPEX independently). We also separated the text for those two points (lines 106-150).

- title of Figure 3 'scientific discovery through active learning' is misleading: the findings are mainly from the entire dataset. Only panel C refers to the learning approach.

Response: We have now changed the title of Figure 3 to “OPEX accelerated the discovery of cluster memberships for the culture conditions”, as Figure 3 is about the clustering results and how OPEX performs at assigning cluster membership. Please see response to the previous point, which further clarifies this issue too. In addition, the updated Fig. 3C is a demonstration of how OPEX accelerates cluster discovery.

We would like to thank the third reviewer for their time and helpful comments.

REFERENCES

- [1]. Fletcher, R. (2013). *Practical methods of optimization*. John Wiley & Sons.
- [2]. Dancik, Garrett M., and Karin S. Dorman. "mleqp: statistical analysis for computer models of biological systems using R." *Bioinformatics* 24.17 (2008): 1966-1967.
- [3]. Gal, Yarin, Riashat Islam, and Zoubin Ghahramani. "Deep bayesian active learning with image data." *Proceedings of the 34th International Conference on Machine Learning-Volume 70*. JMLR. org, 2017.
- [4]. Sener, Ozan, and Silvio Savarese. "Active learning for convolutional neural networks: A core-set approach." *arXiv preprint arXiv:1708.00489* (2017).

Reviewers' Comments:

Reviewer #1:

Remarks to the Author:

Authors have thoroughly revised the document and accounted for most of my comments. I particularly appreciate that the authors provide the code and means of replication of results.

Still, I think that:

1. The abstract and, even better, the title, should clearly indicate that OPEX is developed for "machine learning models".
2. Authors included the reference to the previous work by Sverchkov and Craven (now Ref. 47). However, it is not discussed not mentioned what novelties does OPEX offer with respect to previous alternatives.

I consider this may be easily corrected.

Reviewer #2:

Remarks to the Author:

My concerns/comments have been adequately addressed by the authors.

Reviewer #1 (Remarks to the Author):

Authors have thoroughly revised the document and accounted for most of my comments. I particularly appreciate that the authors provide the code and means of replication of results.

Still, I think that:

1. The abstract and, even better, the title, should clearly indicate that OPEX is developed for "machine learning models".

Response: *Thank you very much for your feedback and time. We have added "using machine learning models" in the abstract to clearly indicate OPEX is based on machine learning models.*

2. Authors included the reference to the previous work by Sverchkov and Craven (now Ref. 47). However, it is not discussed not mentioned what novelties does OPEX offer with respect to previous alternatives. I consider this may be easily corrected.

Response: *We have now expanded the reference to the review by Sverchkov and Craven in Ref. 47. We included the following sentence: "The systems biology applications of OED have been largely focused on uncovering the underlying gene regulatory or signaling network, usually involving a few dozen genes (49,50). In contrast, our focus is on optimal training of predictive machine learning models from genome-scale transcriptional profiling experiments, that aim to capture the expression of thousands of genes.*

Reviewer #2 (Remarks to the Author):

My concerns/comments have been adequately addressed by the authors.